**An improved logistic regression model based on a spatially weighted technique**
**(ILRBSWT v1.0) and its application to mineral prospectivity mapping**
Daojun Zhang[1 2*], Na Ren[1], Xianhui Hou[1*]
[1]College of Economics and Management, Northwest A&F University, Yangling 712100,
China
[2]Center for Resource Economics and Environment Management, Northwest A&F University,
Yangling 712100, China
*Corresponding author: cugzdj@gmail.com (Zhang, D); houxh1019@126.com (Hou, X)
**Abstract:** The combination of complex, multiple minerogenic stages and mineral
superposition during geological processes has resulted in dynamic spatial distributions and
non-stationarity of geological variables. For example, geochemical elements exhibit clear
spatial variability and trends with coverage type changes. Thus, bias is likely to occur under
these conditions when general regression models are applied to mineral prospectivity
mapping (MPM). In this study, we used a spatially weighted technique to improve general
logistic regression and developed an improved model, i.e., the improved logistic regression
model, based on a spatially weighted technique (ILRBSWT, version 1.0). The capabilities and
advantages of ILRBSWT are as follows: (1) it is a geographically weighted regression (GWR)
model, and thus it has all advantages of GWR when managing spatial trends and
non-stationarity; (2) while the current software employed for GWR mainly applies linear
regression, ILRBSWT is based on logistic regression, which is more suitable for MPM
because mineralization is a binary event; (3) a missing data processing method borrowed from
weights of evidence is included in ILRBSWT to extend its adaptability when managing
multisource data; and (4) in addition to geographical distance, the differences in data quality
or exploration level can be weighted in the new model.

**Keywords:** anisotropy; geographical information system modeling; geographically weighted logistic regression; mineral resource assessment; missing data; trend variable; weights of evidence.

**1 Introduction**

The main distinguishing characteristic of spatial statistics compared to classical statistics is that the former has a location attribute. Before geographical information systems were developed, spatial statistical problems were often transformed into general statistical problems, where the spatial coordinates were similar to a sample ID because they only had an indexing feature. However, even in non-spatial statistics, the reversal or amalgamation paradox (Pearson et al., 1899; Yule, 1903; Simpson, 1951), which is commonly called Simpson's paradox (Blyth, 1972), has attracted significant attention from statisticians and other researchers. In spatial statistics, some spatial variables exhibit certain trends and spatial non-stationarity. Thus, it is possible for Simpson's paradox to occur when a classical regression model is applied, and the existence of unknown important variables may worsen this condition. The influence of Simpson's paradox can be fatal. For example, in geology, due to the presence of cover and other factors that occur post-mineralization, ore-forming elements in Area I are much lower than those in Area II, while the actual probability of a mineral in Area I is higher than that in Area II simply because more deposits were discovered in Area I (Agterberg, 1971). In this case, negative correlations would be obtained between ore-forming elements and mineralization according to the classical regression model, whereas high positive correlations can be obtained in both areas if they are separated. Simpson's paradox is an extreme case of bias generated from classical models, and it is usually not so severe in practice. However, this type of bias needs to be considered and care needs to be taken when applying a classical regression model to a spatial problem. Several solutions to

this issue have been proposed, which can be divided into three types.
(1) Locations are introduced as direct or indirect independent variables. This type of
model is still a global model, but space coordinates or distance weights are employed to adjust
the regression estimation between the dependent variable and independent variables
(Agterberg, 1964; Agterberg and Cabilio, 1969; Agterberg, 1970; Agterberg and Kelly, 1971;
Agterberg, 1971; Casetti, 1972; Lesage & Pace, 2009, 2011). For example, Reddy et al. (1991)
performed logistic regression by including trend variables to map the base-metal potential in
the Snow Lake area, Manitoba, Canada; Helbich & Griffith (2016) compared the spatial
expansion method (SEM) to other methods in modeling the house price variation locally,
where the regression parameters are themselves functions of the x and y coordinates and their
combinations; Hao & Liu (2016) used the spatial lag model (SLM) and spatial error model to
control spatial effects in modeling the relationship between $PM_{2.5}$ concentrations and per
capita GDP in China.
(2) Local models are used to replace global models, i.e., geographically weighted models
(Fotheringham et al., 2002). Geographically weighted regression (GWR) is the most popular
model among the geographically weighted models. GWR models were first developed at the
end of the $20^{th}$ century by Brunsdon et al. (1996) and Fotheringham et al. (1996, 1997, 2002)
for modeling spatially heterogeneous processes, and have been used widely in geosciences
(e.g., Buyantuyev & Wu, 2010; Barbet-Massin et al., 2012; Ma et al., 2014; Brauer et al.,

2015).

(3) Reducing trends in spatial variables. For example, Cheng developed a local
singularity analysis technique and spectrum-area (S-A) model based on fractal/multi-fractal
theory (Cheng, 1997; Cheng, 1999). These methods can remove spatial trends and mitigate
the strong effects on predictions of the variables starting at high and low values, and thus they
are used widely to weaken the effect of spatial non-stationarity (e.g., Zhang et al., 2016; Zuo
et al., 2016; Xiao et al., 2017).
GWR models can be readily visualized and are intuitive, which have made them applied
in geography and other disciplines that require spatial data analysis. In general, GWR is a
moving window-based model where instead of establishing a unique and global model for
prediction, it predicts each current location using the surrounding samples, and a higher
weight is given when the sample is located closer. The theoretical foundation of GWR is
Tobler's observation that "everything is related to everything else, but near things are more
related than distant things" (Tobler, 1970).
In mineral prospectivity mapping (MPM), the dependent variables are binary and
logistic regression is used instead of linear regression; therefore, it is necessary to apply
geographically weighted logistic regression (GWLR) instead. GWLR is a type of
geographically weighed generalized linear regression model (Fotheringham et al., 2002) that
is included in the software module GWR 4.09 (Nakaya, 2016). However, the function module
for GWLR in current software can only manage data in the form of a tabular dataset
containing the fields with dependent and independent variables and x-y coordinates.
Therefore, the spatial layers have to be re-processed into two-dimensional tables and the
resulting data needs to be transformed back into a spatial form.
Another problem with applying GWR 4.09 for MPM is that it cannot handle missing
data (Nakaya, 2016). Weights of evidence (WofE) is a widely used model for MPM
(Bonham-Carter et al., 1988, 1989; Agterberg, 1989; Agterberg et al., 1990) that mitigates the
effects of missing data. However, WofE was developed assuming that conditional
independence is satisfied among evidential layers with respect to the target layer; otherwise,
the posterior probabilities will be biased, and the number of estimated deposits will be
unequal to the known deposits. Agterberg (2011) combined WofE with logistic regression and
proposed a new model that can obtain an unbiased estimate of number of deposits while also
avoiding the effect of missing data. In this study, we employed Agterberg (2011) 's solution to
account for missing data.

One more improvement of the ILRBSWT is that a mask layer is included in the new

model to address data quality and exploration level differences between samples.

Conceptually, this research originated from the thesis of Zhang (2015; in Chinese),

which showed better efficiency for mapping intermediate and felsic igneous rocks (Zhang
et al., 2017). This work elaborates on the principles of ILRBSWT, and provides a detailed
algorithm for its design and implementation process with the code and software module
attached. In addition, processing missing data is not a technique covered in GWR
modeling presented in prior research, and a solution borrowed from WofE is provided in
this study. Finally, ILRBSWT performance in MPM is tested by predicting Au ore
deposits in western Meguma Terrain, Nova Scotia, Canada.

**2 Models**
Linear regression is commonly used for exploring the relationship between a response
variable and one or more explanatory variables. However, in MPM and other fields, the
response variable is binary or dichotomous, so linear regression is not applicable and thus a
logistic model is advantageous.
*2.1 Logistic Regression*
In MPM, the dependent variable($Y$) is binary because $Y$ can only take the value of 1 and 0,
indicating that mineralization occurs and not respectively. Suppose that $\pi$ represents the
estimation of $Y$, $0 \leqslant \pi \leqslant 1$, then a logit transformation of $\pi$ can be made, i.e., logit ($\pi$)
$=\ln(\pi/(1-\pi))$. The logistic regression function can be obtained as follows:
Logit $\pi\left(X_1, X_2, \cdots, X_p\right) = \beta_0 + \beta_1 X_1 + \cdots + \beta_p X_p$                     (1)
where $X_1, X_2, \cdots, X_p$, comprises a sample of $p$ explanatory variables $x_1, x_2, \cdots, x_p$, $\beta_0$ is the
intercept, and $\beta_1, \beta_2, \cdots, \beta_p$ are regression coefficients.

If there are $n$ samples, we can obtain $n$ linear equations with $p+1$ unknowns based on

equation (1). Furthermore, if we suppose that the observed values for $Y$ are $Y_1, Y_2, \cdots, Y_n$, and
these observations are independent of each other, then a likelihood function can be
established:
$L(\beta) = \prod_{i=1}^{n}(\pi_i^{Y_i}(1-\pi_i)^{1-Y_i}),$             (2)
where $\pi_i = \pi(X_{i1}, X_{i2}, \cdots, X_{ip}) = \frac{e^{\beta_0 + \beta_1 X_{i1} + \cdots + \beta_p X_{ip}}}{1 + e^{\beta_0 + \beta_1 X_{i1} + \cdots + \beta_p X_{ip}}}$. The best estimate can be obtained
only if equation (2) takes the maximum. Then the problem is converted into solving
$\beta_1, \beta_2, \cdots, \beta_p$. Equation (2) can be further transformed into the following log-likelihood
function:
$\ln L(\beta) = \sum_{i=1}^{n}(Y_i\pi_i + (1-Y_i)(1-\pi_i))$             (3)

The solution can be obtained by taking the first partial derivative of $\beta_i$ ($i = 0$ to $p$),

which should be equal to 0:
$$\begin{cases} f(\beta_0) = \sum_{i=0}^{n}(Y_i - \pi_i)X_{i0} = 0 \\ f(\beta_1) = \sum_{i=0}^{n}(Y_i - \pi_i)X_{i1} = 0 \\ \quad\quad\quad\vdots \\ f(\beta_p) = \sum_{i=0}^{n}(Y_i - \pi_i)X_{ip} = 0 \end{cases}$$             (4)
where $X_{i0} = 1$, $i$ takes the value from 1 to $n$, and equation (4) is obtained in the form of
matrix operations.
$\mathbf{X}^\mathbf{T}(\mathbf{Y} - \mathbf{\pi}) = \mathbf{0}$             (5)

The Newton iterative method can be used to solve the nonlinear equations:

$\hat{\mathbf{\beta}}(t+1) = \hat{\mathbf{\beta}}(t) + \mathbf{H}^{-1}\mathbf{U}$ ,             (6)
where $\mathbf{H} = \mathbf{X}^\mathbf{T}\mathbf{V}(t)\mathbf{X}$, $\mathbf{U} = \mathbf{X}^\mathbf{T}(\mathbf{Y} - \mathbf{\pi}(t))$, $t$ represents the number of iterations, and $\mathbf{V}(t)$, $\mathbf{X}$,
$\mathbf{Y}$, $\mathbf{\pi}(t)$, and $\hat{\mathbf{\beta}}(t)$ are obtained as follows:
$$\mathbf{V}(t) = \begin{pmatrix} \pi_1(t)(1-\pi_1(t)) & & & \\ & \pi_2(t)(1-\pi_2(t)) & & \\ & & \ddots & \\ & & & \pi_n(t)(1-\pi_n(t)) \end{pmatrix},$$
$$\mathbf{X} = \begin{pmatrix} X_{10} & X_{11} & \cdots & X_{1p} \\ X_{20} & X_{21} & \cdots & X_{2p} \\ \vdots & \vdots & \ddots & \vdots \\ X_{n0} & X_{n1} & \cdots & X_{np} \end{pmatrix}, \ \mathbf{Y} = \begin{pmatrix} Y_1 \\ Y_1 \\ \vdots \\ Y_n \end{pmatrix}, \ \boldsymbol{\pi}(t) = \begin{pmatrix} \pi_1(t) \\ \pi_2(t) \\ \vdots \\ \pi_n(t) \end{pmatrix}, \text{ and } \ \widehat{\boldsymbol{\beta}}(t) = \begin{pmatrix} \hat{\beta}_1(t) \\ \hat{\beta}_2(t) \\ \vdots \\ \hat{\beta}_n(t) \end{pmatrix}.$$
For a more detailed description of the derivations of equations (1) to (6), see Hosmer et al.

(2013).

*2.2 Weighted Logistic Regression*
In practice, vector data is often used, and sample size (area) has to be considered. In this
condition, weighted logistic regression modeling should be used instead of a general logistic
regression. It is also preferable to use a weighted logistic regression model when a logical
regression should be performed for large sample data because weighted logical regression can
significantly reduce matrix size and improve computational efficiency (Agterberg, 1992).
Assuming that there are four binary explanatory variable layers and the study area consists of
1000×1000 grid points, the matrix size for normal logic regression modeling would be
$10^6 \times 10^6$; however, if weighted logistic regression is used, the matrix size would be 32×32 at
most. This condition arises because the sample classification process is contained in the
weighted logistic regression, and all samples are classified into classes with the same values
as the dependent and independent variables. The samples with the same dependent and
independent variables form certain continuous and discontinuous patterns in space, which are
called "unique condition" units. Each unique condition unit is then treated as a sample, and
the area (grid number) for it is taken as weight in the weighed logistic regression. Thus, for
the weighted logical regression, equations (2) to (5) in section 2.1 need to be changed to
equations (7) to (10) as follows.

$L_{new}(\beta) = \prod_{i=1}^{n}(\pi_i^{N_iY_i}(1-\pi_i)^{N_i(1-Y_i)})$,                                        (7)
$\ln L_{new}(\beta) = \sum_{i=1}^{n}(N_iY_i\pi_i + N_i(1-Y_i)(1-\pi_i))$                      (8)
$\begin{cases} f_{new}(\beta_0) = \sum_{i=0}^{n}(Y_i-\pi_i)X_{i0} = 0 \\ f_{new}(\beta_1) = \sum_{i=0}^{n}(Y_i-\pi_i)X_{i1} = 0 \\ \qquad\qquad \vdots \\ f_{new}(\beta_p) = \sum_{i=0}^{n}(Y_i-\pi_i)X_{ip} = 0 \end{cases}$            (9)
$\mathbf{X^TW(Y-\pi) = 0}$                                                   (10)
where $N_i$ is the weight for the $i$-th unique condition unit, $i$ takes the value from 1 to $n$, and $n$
is the number of unique condition units. $\mathbf{W}$ is a diagonal matrix that is expressed as follows:

$$\mathbf{W} = \begin{pmatrix} N_1 & & & \\ & N_2 & & \\ & & \ddots & \\ & & & N_n \end{pmatrix}$$

In addition, new values of H and U should be used in equation (6) to perform Newton
iteration as part of the weighted logistic regression, i.e., $\mathbf{H}_{new} = \mathbf{X^TWV}(t)\mathbf{X}$, $\mathbf{U}_{new} =$
$\mathbf{X^TW(Y-\pi}(t))$.
*2.3 Geographically Weighted Logistic Regression*
GWLR is a local window-based model where logistic regression is established at each current
location in the GWLR. The current location is changed using the moving window technique
with a loop program. Suppose that $\mathbf{u}$ represents the current location, which can be uniquely
determined by a pair of column and row numbers, $x$ denotes $p$ explanatory variables
$x_1, x_2, \cdots, x_p$ that take values of $X_1, X_2, \cdots, X_p$ respectively, and $\pi(x, \mathbf{u})$ is the $Y$ estimate, i.e.,
the probability that $Y$ takes a value of 1, and then the following function can be obtained.
$\text{Logit } \pi(x, \mathbf{u}) = \beta_0(\mathbf{u}) + \beta_1(\mathbf{u})x_1 + \beta_2(\mathbf{u})x_2 + \cdots + \beta_p(\mathbf{u})x_p$ ,             (11)
where, $\beta_0(\mathbf{u})$, $\beta_1(\mathbf{u})$, $\cdots$, $\beta_p(\mathbf{u})$ indicate that these parameters are obtained at the location
of $\mathbf{u}$. $\text{Logit } \pi(x, \mathbf{u})$, the predicted probability for the current location $\mathbf{u}$, can be obtained under
the condition that the values of all independent variables are known at the current location and
all parameters are also calculated based on the samples within the current local window.
According to equation (6) in section 2.1, the parameters for GWLR can be estimated with
equation (12):
$\hat{\boldsymbol{\beta}}(\mathbf{u})_{t+1} = \hat{\boldsymbol{\beta}}(\mathbf{u})_t + (\mathbf{X}^T\mathbf{W}(\mathbf{u})\mathbf{V}(t)\mathbf{X})^{-1}\mathbf{X}^T\mathbf{W}(\mathbf{u})(\mathbf{Y} - \boldsymbol{\pi}(t)),$           (12)
where $t$ represents the number of iterations; $\mathbf{X}$ is a matrix that includes the values of all
independent variables, and all elements in the first column are 1; $\mathbf{W}(\mathbf{u})$ is a diagonal matrix
where the diagonal elements are geographical weights, which can be calculated according to
distance, whereas the other elements are all 0; $\mathbf{V}(t)$ is also a diagonal matrix and the
diagonal element can be expressed as $\pi_i(t)(1 - \pi_i(t))$; and $\mathbf{Y}$ is a column vector
representing the values taken by the dependent variable.
*2.4 Improved Logistic Regression Model based on the Spatially Weighted Technique*
As is mentioned in the introduction section, there are primarily two improvements for
ILRBSWT compared to GWLR, i.e., the capacity to manage different types of weights, and
the special handling of missing data.
*2.4.1 Integration of Different Weights*
If a diagonal element in $\mathbf{W}(\mathbf{u})$ is only for one sample, i.e., the grid point in raster data, section
2.3 is an improvement on section 2.1, i.e. samples are weighted according to their location. If
samples are first reclassified according to the unique condition mentioned in section 2.2, and
corresponding weights are then summarized according to each sample's geographical weight,
we can obtain an improved logistic regression model considering both sample size and
geographical distance. The new model both reflects the spatial distribution of samples and
reduces the matrix size, which is discussed in the following section.
In addition to geographic factors, representativeness of a sample, e.g., differences in the
level of exploration, is also considered in this study.
Suppose that there are $n$ grid points in the current local window, $S_i$ is the $i$-th grid, $W_i(g)$
is the geographical weight of $S_i$, and $W_i(d)$ represents the individual difference weight or
non-geographical weight. In some cases, there may be differences in quality or the exploration
level among samples, but $W_i(d)$ takes a value of 1 if there is no difference, where $i$ takes a
value from 1 to $n$. Furthermore, if we suppose that there are $N$ unique conditions after
overlaying all layers ($N \leq n$) and $C_j$ denotes the $j$-th unique condition unit, then we can obtain
the final weight for each unique condition unit in the current local window:
$$W_j(t) = \sum_{i=1}^{n}[W_i(\text{g}) * W_i(d) * \text{df}_i], \tag{13}$$
where $\begin{cases} \text{df}_i = 1 & \text{if } S_i \in C_j \\ \text{df}_i = 0 & \text{if } S_i \notin C_j \end{cases}$, $i$ takes a value from 1 to $n$, $j$ takes a value from 1 to $N$, and $W_j(t)$
represents the total weight (by combining both $W_i(g)$ and $W_i(d)$) for each unique condition
unit. We can use the final weight calculated in equation (13) to replace the original weight in
equation (12), which is an advantage of ILRBSWT.
*2.4.2 Missing data processing*
Missing data is a problem in all statistics-related research fields. In MPM, missing data are
also prevalent due to ground coverage, and limitations of exploration technique and
measurement accuracy. Agterberg and Bonham-Carter (1999) compared the following
commonly used missing data processing solutions: (1) removing variables containing missing
data, (2) deleting samples with missing data, (3) using 0 to replace missing data, and (4)
replacing missing data with the mean of the corresponding variable. To efficiently use
existing data, both (1) and (2) are clearly not good solutions as more data will be lost.
Solution (3) is superior to (4) in the condition that work has not been done and real data is
unknown; with respect to missing data caused by detection limits, solution (4) is clearly a
better choice. Missing data is primarily caused by the latter in MPM, and Agterberg (2011)
pointed out that missing data was better addressed in a WofE model. In WofE, the evidential
variable takes the value of positive weight ($W^+$) if it is favorable for the target variable (e.g.,
mineralization), while the evidential variable takes the value of negative weight ($W^-$) if it is
unfavorable for the target variable; and automatically the evidential variable takes the value of
"0" if there is missing data.
$$W^+ = \ln \frac{\frac{D_1}{D}}{\frac{A_1 - D_1}{A - D}}$$     (14)
$$W^- = \ln \frac{\frac{D_2}{D}}{\frac{A_2 - D_2}{A - D}}$$     (15)
where $\mathbf{A}$ is an evidential layer, $A_1$ is the area (or grid number, similarly hereinafter) that $A$
takes the value of 1, and $A_2$ is the area that $A$ takes the value of 0; $A_3$ is the area with missing
data, and $A_1 + A_2$ is smaller than the total study area if missing data exists. $D_1$, $D_2$, and $D_3$ are
areas where the target variables take the value of 1 in $A_1$, $A_2$, and $A_3$ respectively. $A_3$ and $D_3$
are not used in equation (15) because no information is provided in area $A_3$.

If "1" and "0" are still used to represent the binary condition of the independent variable

instead of $W^+$ and $W^-$, equation (16) can be used to replace missing data in logistic
regression modeling.
$$M = \frac{-W^-}{W^+ - W^-} = \frac{\ln\frac{D}{A-D} - \ln\frac{D_2}{A_2 - D_2}}{\ln\frac{D_1}{A_1 - D_1} - \ln\frac{D_2}{A_2 - D_2}}$$     (16)

**3 Design of the ILRBSWT Algorithm**
*3.1 Local Window Design*
A raster data set is used for ILRBSWT modeling. With regular grids, the distance between any
two grid points can be calculated easily and distance templates within a certain window scope
can be obtained, which is highly efficient for data processing. The circle and ellipse are used
for isotropic and anisotropic local window designs, respectively.

(1) Circular Local Window Design

Suppose that $W$ represents a local circular window where the minimum bounding

rectangle is $R$, then the geographical weights can be calculated only inside $R$. Clearly, the grid
points inside $R$ but outside of $W$ should be weighted as 0, and the weight for the grid with a
center inside $W$ should be calculated according to the distance from its current location.
Because $R$ is a square, we can also assume that there are $n$ columns and rows in it, where $n$ is
an odd number. If we take east and south as the orientations of the $x$-axis and $y$-axis,
respectively, and the position of the northwest corner grid is defined as $(x = 1, y = 1)$, then a
local rectangular coordinate system can be established and the position of the current location
grid can be expressed as $O\ (x = \frac{n+1}{2},\ y = \frac{n+1}{2})$. The distance between any grid inside $W$ and
the current location grid can be expressed as $d_{o-ij} = \sqrt{\left(i - \frac{n+1}{2}\right)^2 + \left(j - \frac{n+1}{2}\right)^2}$, where $i$ and
$j$ take values ranging from 1 to $n$. The geographical weight is a function of distance, so it is
convenient to calculate $w_{ij}$ with $d_{o-ij}$. Figure 1 shows the weight template for a circular
local window with a half-window size of nine grids.

| | | | | | | | | | | | | | | | | |
|---|---|---|---|---|---|---|---|---|---|---|---|---|---|---|---|---|
| 0 | 0 | 0 | 0 | 0 | 0 | 0 | 0 | w30 | 0 | 0 | 0 | 0 | 0 | 0 | 0 | 0 |
| 0 | 0 | 0 | 0 | 0 | w28 | w27 | w25 | w24 | w25 | w27 | w28 | 0 | 0 | 0 | 0 | 0 |
| 0 | 0 | 0 | w29 | w26 | w23 | w21 | w20 | w19 | w20 | w21 | w23 | w26 | w29 | 0 | 0 | 0 |
| 0 | 0 | w29 | w25 | w22 | w18 | w16 | w15 | w14 | w15 | w16 | w18 | w22 | w25 | w29 | 0 | 0 |
| 0 | 0 | w26 | w22 | w17 | w14 | w13 | w11 | w10 | w11 | w13 | w14 | w17 | w22 | w26 | 0 | 0 |
| 0 | w28 | w23 | w18 | w14 | w12 | w9 | w8 | w7 | w8 | w9 | w12 | w14 | w18 | w23 | w28 | 0 |
| 0 | w27 | w21 | w16 | w13 | w9 | w6 | w5 | w4 | w5 | w6 | w9 | w13 | w16 | w21 | w27 | 0 |
| 0 | w25 | w20 | w15 | w11 | w8 | w5 | w3 | w2 | w3 | w5 | w8 | w11 | w15 | w20 | w25 | 0 |
| w30 | w24 | w19 | w14 | w10 | w7 | w4 | w2 | w1 | w2 | w4 | w7 | w10 | w14 | w19 | w24 | w30 |
| 0 | w25 | w20 | w15 | w11 | w8 | w5 | w3 | w2 | w3 | w5 | w8 | w11 | w15 | w20 | w25 | 0 |
| 0 | w27 | w21 | w16 | w13 | w9 | w6 | w5 | w4 | w5 | w6 | w9 | w13 | w16 | w21 | w27 | 0 |
| 0 | w28 | w23 | w18 | w14 | w12 | w9 | w8 | w7 | w8 | w9 | w12 | w14 | w18 | w23 | w28 | 0 |
| 0 | 0 | w26 | w22 | w17 | w14 | w13 | w11 | w10 | w11 | w13 | w14 | w17 | w22 | w26 | 0 | 0 |
| 0 | 0 | w29 | w25 | w22 | w18 | w16 | w15 | w14 | w15 | w16 | w18 | w22 | w25 | w29 | 0 | 0 |
| 0 | 0 | 0 | w29 | w26 | w23 | w21 | w20 | w19 | w20 | w21 | w23 | w26 | w29 | 0 | 0 | 0 |
| 0 | 0 | 0 | 0 | 0 | w28 | w27 | w25 | w24 | w25 | w27 | w28 | 0 | 0 | 0 | 0 | 0 |
| 0 | 0 | 0 | 0 | 0 | 0 | 0 | 0 | w30 | 0 | 0 | 0 | 0 | 0 | 0 | 0 | 0 |


**Fig. 1 Weight template for a circular local window with a half-window size of nine grids, where w1 to w30 represent different weight classes that decrease with distances and 0 indicates that the grid is weighted as 0. Gradient colors ranging from red to green are used to distinguish the weight classes for grid points.**

Suppose that there are $T\_n$ columns and $T\_m$ rows in the study area, and *Current* ($T\_i$, $T\_j$) represents the current location, where $T\_i$ takes values from 1 to $T\_n$ and $T\_j$ takes values from 1 to $T\_m$, then the current local window can be established by selecting the range of rows $T\_i - \frac{n-1}{2}$ to $T\_i + \frac{n-1}{2}$ and columns $T\_j - \frac{n-1}{2}$ to $T\_j + \frac{n-1}{2}$ from the total research area. Next, we can establish a local rectangular coordinate system according to the previously described steps; we subtract $T\_i - \frac{n-1}{2}$ and $T_j - \frac{n-1}{2}$ on the *x* and *y* coordinates respectively for all grids in the range. The corresponding relationship can then be established

between the weight template and current window. Global weights can also be included via the
matrix product between the global weight layer and local weight template within the local
window. In addition, special care should be taken when the weight template covers some area
outside the study area, i.e., $T\_i - \frac{n-1}{2} < 0$, $T\_i + \frac{n-1}{2} > T\_n$, $T\_j - \frac{n-1}{2} < 0$, and $T\_j +$
$\frac{n-1}{2} > T\_m$.
(2) Elliptic Local Window Design
In most cases, the spatial tendency of the spatial variable may vary with different
directions and an elliptic local window may better describe the changes in weights in space.
To simplify the calculation, we can convert the distances in different directions into equivalent
distances, and an anisotropic problem is then converted into an isotropic problem. For any
grid, the equivalent distance is the semi-major axis length of the ellipse that is centered at the
current location and passes through the grid, while the parameters for the ellipse can be
determined using the kriging method.
We still use $W$ to represent the local elliptic window and $a$, $r$, and $\theta$ are defined as the
semi-major axis, ratio of the semi-minor axis relative to the semi-major axis, and azimuth of
the semi-major axis, respectively. Then, $W$ can be covered by a square $R$ whose side length is
$2a$-1 and center is the same as $W$. There are $(2a - 1) \times (2a - 1)$ grids in $R$. We establish the
rectangular coordinates as described above and suppose that the center of the top left grid in $R$
is located at $(x = 1, y = 1)$, and thus the center of $W$ should be $O(x_0 = a, y_0 = a)$. According
to the definition of the ellipse, two of the elliptical focuses are located at $F_1\big(x_1 = a +$
$\sin(\theta)\sqrt{a^2 - (a * r)^2}, y_1 = a - \mathrm{con}(\theta)\sqrt{a^2 - (a * r)^2}\big)$     and
$F_2 \ (x_2 = a - \sin(\theta)\sqrt{a^2 - (a * r)^2}, y_2 = a + \mathrm{con}(\theta)\sqrt{a^2 - (a * r)^2})$ .    The    summed
distances    between    a    point    and    two    focus    points    can    be    expressed    as
$l_{ij} = \sqrt{(i - x_1)^2 + (j - y_1)^2} + \sqrt{(i - x_2)^2 + (j - y_2)^2}$, where $i$ and $j$ take values from 1 to
$2a - 1$. According to the elliptical focus equation, for any grid in $R$, if the sum of the distances
between the two focal points and a grid center is greater than $2a$, the grid is located within $W$,
and vice versa. For the grids outside of $W$, the weight is assigned as 0, and we only need to
calculate the equivalent distances for the grids within $W$. As mentioned above, the parameters
for the ellipse can be determined using the kriging method. In ellipse $W$, where the semi-major
axis is $a$, $r$ and $\theta$ are maintained as constants, then we obtain countless ellipses centered at
the center of $W$, and the equivalent distance is the same on the same elliptical orbit. Thus, the
equivalent distance template can be obtained for the local elliptic window. Figure 2 shows the
equivalent distance templates under the conditions that $a = 11$ grids, $r = 0.5$, and the azimuths
for the semi-major axis are 0°, 45°, 90°, and 135° respectively.

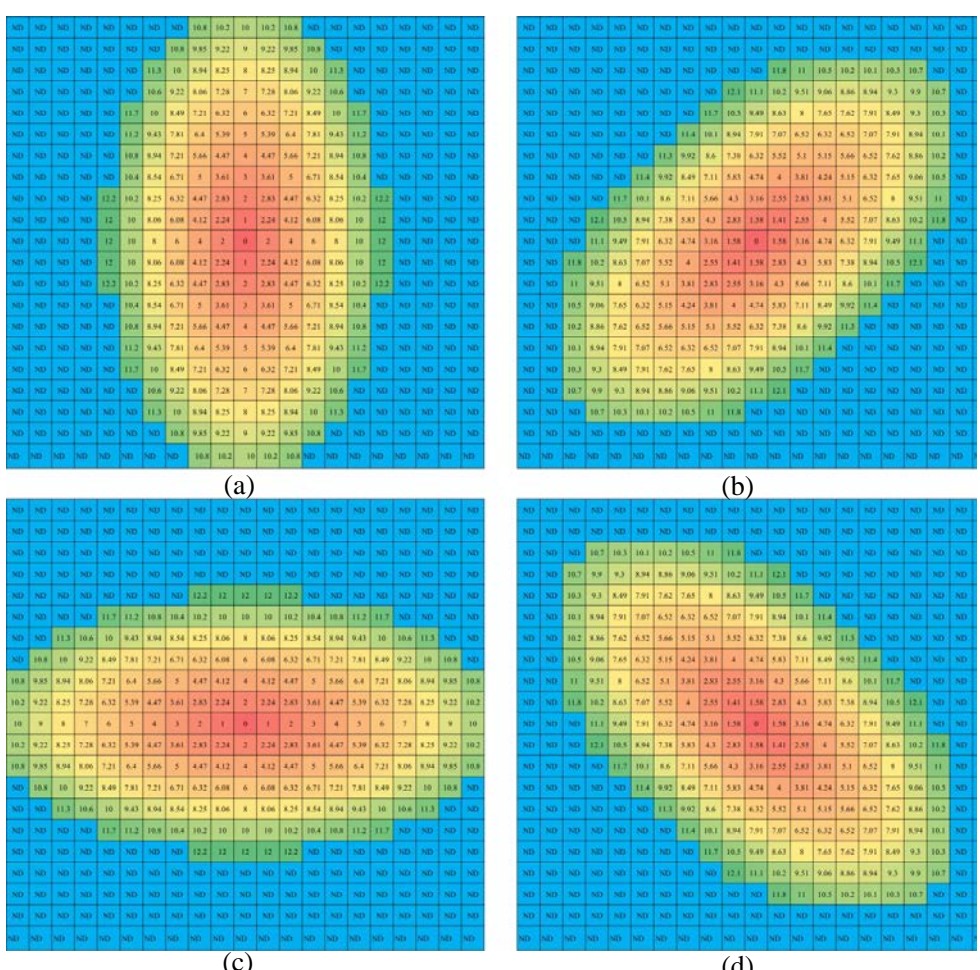

(c)                                    (d)

**Fig. 2 Construction of the distance template based on an elliptic local window: $a = 11$ grid points,**
**$r = 0.5$, and the azimuths for the semi-major axis are 0° (a), 45°(b), 90° (c), and 135° (d) respectively.**
*3.2 Algorithm for ILRBSWT*
The ILRBSWT method primarily focuses on two problems, i.e., spatial non-stationarity and
missing data. We use the moving window technique to establish local models instead of a
global model to overcome spatial non-stationarity. The spatial *t*-value employed in the WofE
method is used to binarize spatial variables based on the local window, which is quite
different from traditional binarization based on the global range, where missing data can be
handled well because positive and negative weights are used instead of the original values of
"1" and "0", and missing data are represented as "0." Both the isotropy and anisotropy
window types are provided in our new proposed model. The geographical weight function and
window size can be determined by the users. If the geographic weights are equal and there are
no missing data, ILRBSWT will yield the same posterior probabilities as classical logistic
regression; hence, the later can be viewed as a special case of the former. The core ILRBSWT
algorithm is as follows.
Step 1. Establish a loop for all grids in the study area according to both the columns and
rows. Determine a basic local window with a size of $r_{\min}$ based on a variation function or
other method. In addition, the maximum local window size is set as $r_{\max}$, with an interval of
$\Delta R$. Suppose that a geographical weighted model has already been given in the form of a
Gaussian curve determined from variations in geostatistics, i.e., $W(g) = e^{-\lambda d^2}$, where $d$ is
the distance and $\lambda$ is the attenuation coefficient, then we can calculate the geographical
weight for any grid in the current local window. The equivalent radius should be used in the
anisotropic situation. When other types of weights are considered, e.g., the degree of
exploration or research, it is also necessary to synthesize the geographical weights with other
weights (see equation 13).
Step 2. Establish a loop for all independent variables. In a circular (elliptical) window
with a radius (equivalent radius) of $r_{\min}$, apply the WofE (Agterberg, 1992) model according
to the grid weight determined in step 1, thereby obtaining a statistical table containing the
parameters $W_{ij}^{+}$, $W_{ij}^{-}$, and $t_{ij}$, where $i$ is the $i$-th independent variable and $j$ denotes the $j$-th
binarization.
Step 2.1. If a maximum $t_{ij}$ exists and it is greater than or equal to the standard $t$-value
(e.g., 1.96), record the values of $W_{i-\max\_t}^{+}$, $W_{i-\max\_t}^{-}$, and $B_{i-\max\_t}$, which denote the
positive weight, negative weight, and corresponding binarization, respectively, under the
condition where $t$ takes the maximum value. Go to step 2 and apply the WofE model to the
other independent variables.
Step 2.2. If a maximum $t_{ij}$ does not exist, or it is smaller than the standard $t$-value, go to
step 3.
Step 3. In a circular (elliptical) window with a radius (equivalent radius) of $r_{\max}$, increase
the current local window radius from $r_{\min}$ according to the algorithm in step 1.
Step 3.1. If all independent variables have already been processed, go to step 4.
Step 3.2. If the size of the current local window exceeds the size of $r_{\max}$, disregard the
current independent variable and go to step 2 to consider the remaining independent variables.
Step 3.3. Apply the WofE model according to the grid weight determined in step 1 in the
current local window. If a maximum $t_{ij}$ exists and it is greater than or equal to the standard
$t$-value, record the values of $W_{i-\max\_t}^{+}$, $W_{i-\max\_t}^{-}$, $B_{i-\max\_t}$, and $r_{\text{current}}$, which represent the
radius (equivalent radius) for the current local window.
Step 3.4. If a maximum $t_{ij}$ does not exist or it is smaller than the standard $t$-value, go to
step 3.
Step 4. Suppose that $n_s$ independent variables still remain.
Step 4.1. If $n_s \leq 1$, calculate the mean value for the dependent variable in the current
local window with a radius size of $r_{\max}$ and retain it as the posterior probability in the current
location. In addition, set the regression coefficients for all independent variables as missing
data. Go to step 6.

Step 4.2. If $n_s \geq 1$, find the independent variable with the largest local window and

apply the WofE model to all other independent variables, and then update the values of
$W^+_{i-\max\_t}$, $W^-_{i-\max\_t}$, and $B_{i-\max\_t}$. Go to step 5.

Step 5. Apply the logistic regression model based on the previously determined

geographic weights, and for each independent variable: (1) use $W^+_{i-\max\_t}$ to replace all
values that are less than or equal to $B_{i-\max\_t}$, (2) use $W^-_{i-\max\_t}$ to replace all values that are
greater than $B_{i-\max\_t}$, and (3) use 0 to replace no data ("-9999"). The posterior probability
and regression coefficients can then be obtained for all independent variables at the current
location and go to step 6.

Step 6. Take the next grid as the current location and repeat steps 2–5.


**4 Interface Design**
Before performing spatially weighted logistical regression with ILRBSWT 1.0, data
pre-processing is performed using the ArcGIS 10.2 platform and GeoDAS 4.0 software. All
data are originally stored in grid format, which should be transformed into ASCII files with
the Arc toolbox in ArcGIS 10.2; after modeling with ILRBSWT 1.0, the result data will be
transformed back into grid format

As shown in Fig. 3, the main interface for ILRBSWT 1.0 is composed of four parts.

The upper left part is for the layer input settings, where independent variable layers,

dependent variable layers, and global weight layers should be assigned. Layer information is
shown at the upper right corner, including row numbers, column numbers, grid size, ordinate
origin, and the expression for missing data. The local window parameters and weight
attenuation function can be defined as follows. Using the drop-down list, we prepared a circle

397 or ellipse to represent various isotropic and anisotropic spatial conditions, respectively. The

398 corresponding window parameters should be set for each window type. For the ellipse, it is

399 necessary to set parameters composed of the initial length of the equivalent radius (initial

400 major radius), final length of the equivalent radius (largest major radius), increase in the

401 length of the equivalent radius (growth rate), threshold of the spatial $t$-value used to determine

402 the need to enlarge the window, length ratio of the major and minor axes, orientation of the

403 ellipse's major axis, and compensation coefficient for the sill. We prepared different types of

404 weight attenuation functions via the drop-down menu to provide choices to users, such as

405 exponential model, logarithmic model, Gaussian model, and spherical model, and

406 corresponding parameters can be set when a certain model is selected. The output file is

407 defined at the bottom and the execution button is at the lower right corner.

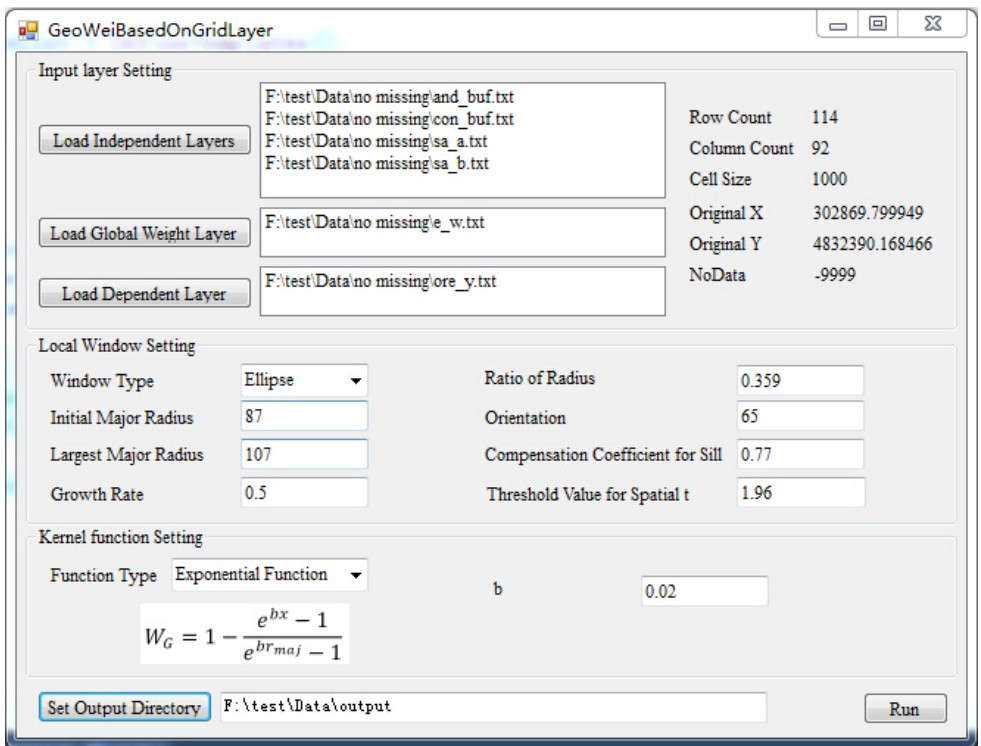


409         **Fig. 3 User interface design.**




## 5 Real Data Testing

*5.1 Data source and preprocessing*

The test data used in this study were obtained from the case study reported in Cheng (2008). The study area ($\approx 7780$ km$^2$) is located in western Meguma Terrain, Nova Scotia, Canada. Four independent variables were used in the WofE model for gold mineral potential mapping by Cheng (2008), i.e., buffer of anticline axes, buffer for the contact of Goldenville–Halifax Formation, and background and anomaly separated with the S-A filtering method based on ore element loadings of the first component, as shown in Fig. 4.

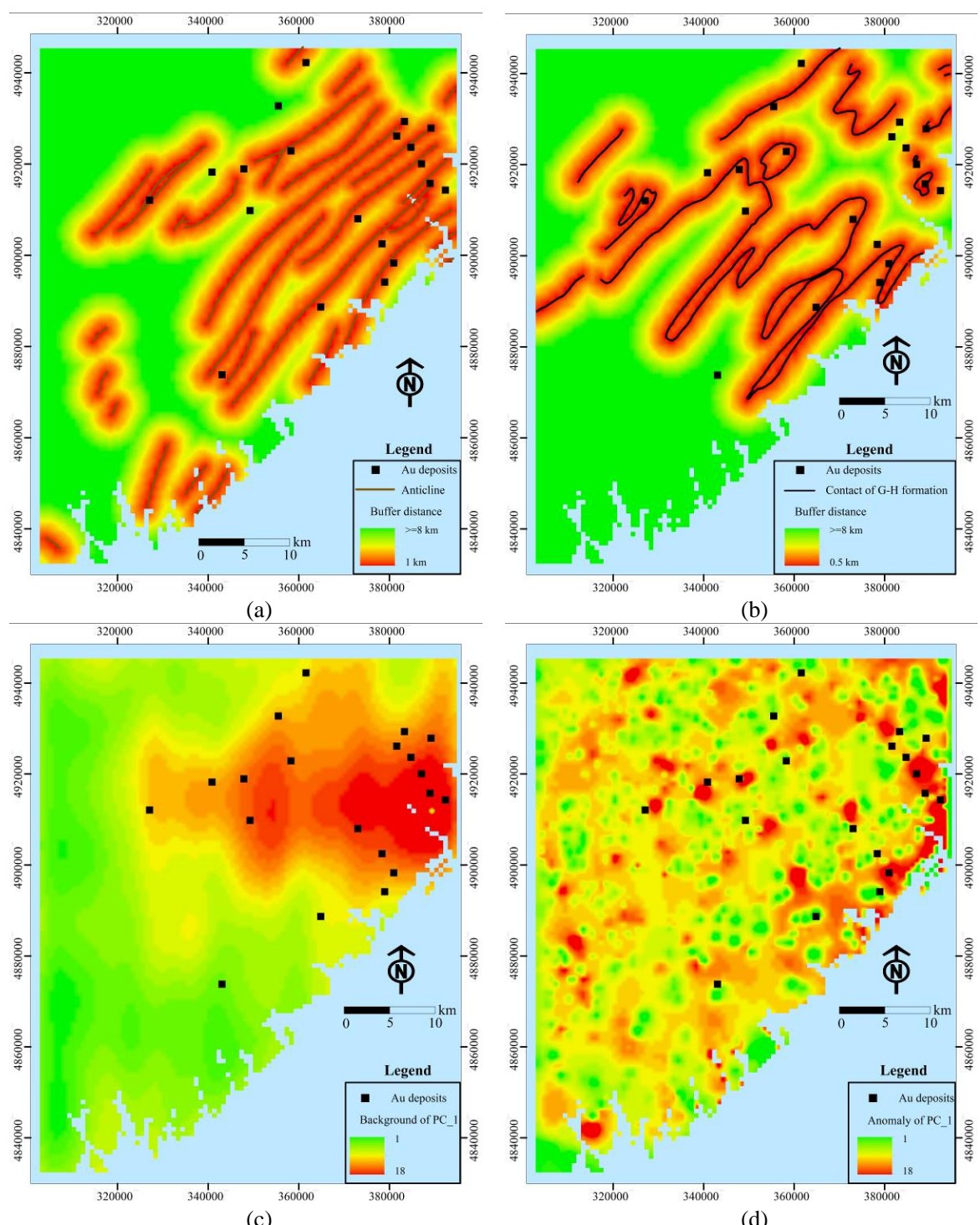

**Fig. 4 Evidential layers used to map Au deposits in this study: buffer of anticline axes (a), buffer for**

**the contact of Goldenville–Halifax Formation (b), and background (c) and anomaly (d) separated**

**with the S-A filtering method based on the ore element loadings of the first component.**

The four independent variables described previously were also used for ILRBSWT

modeling in this study (see Figs. 4 (a) to (d)), and they were uniformed in the ArcGIS grid

format with a cell size of 1 km × 1 km. To demonstrate the advantages of the new method for

missing data processing, we designed an artificial situation in Fig. 5, i.e., grids in region A
have values for all four independent variables, while they only have values for two
independent variables and no values in the two geochemical variables in region B.

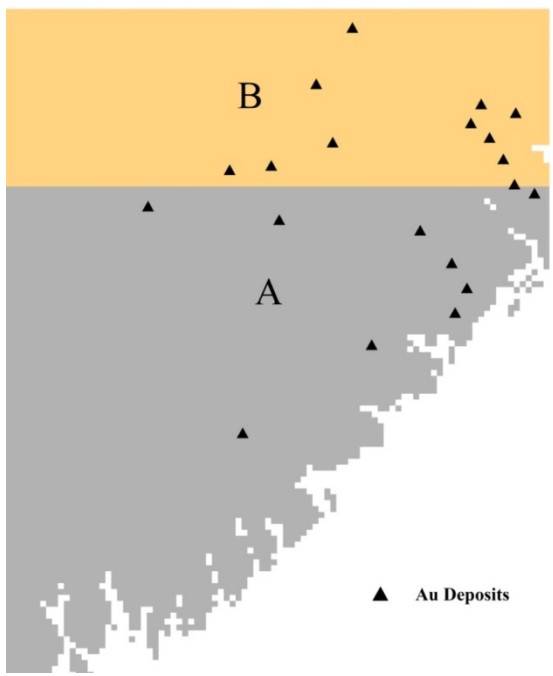


**Fig. 5 Study area (A and B) where there is missing geochemical data in area B.**
*5.2 Mapping weights for exploration*
Exploration level weights can be determined based on prior knowledge about data quality, e.g.,
different scales may exist throughout the whole study area; however, these weights can also
be calculated quantitatively. The density of known deposits is a good index for the exploration
level, i.e., the research is more comprehensive when more deposits are discovered. The
exploration level weight layer for the study area was obtained using the kernel density tool
provided by the ArcToolbox in ArcGIS 10.2, as shown in Fig. 6.

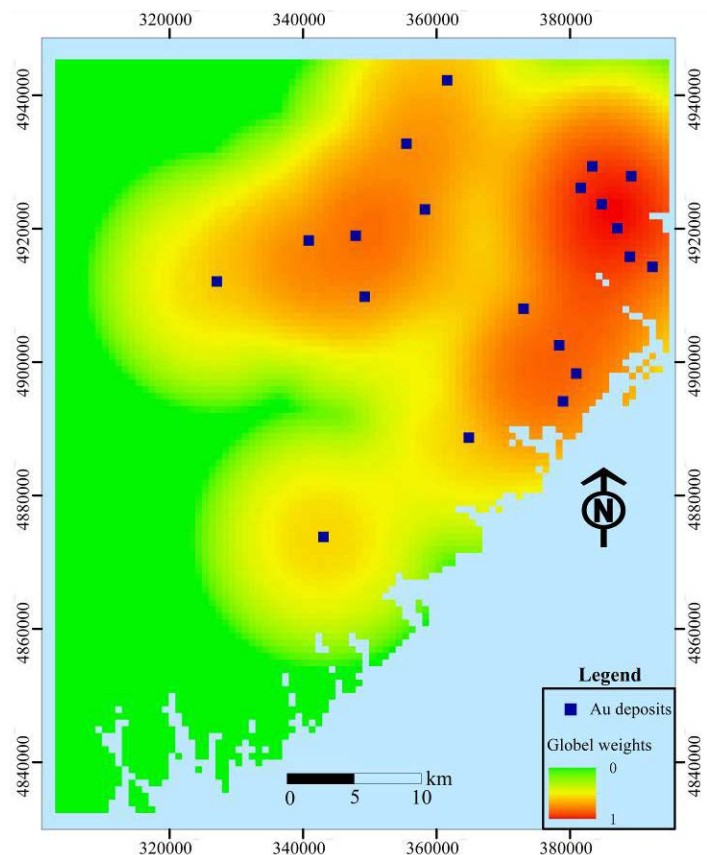

**Fig. 6 Exploration level weights.**

*5.3 Parameter Assignment for local window and weight attenuation function*

Both empirical and quantitative methods can be used to determine the local window parameters and attenuation function for geographical weights. The variation function in geostatistics, which is an effective method for describing the structures and trends in spatial variables, was applied in this study. To calculate the variation function for the dependent variable, it is necessary to first map the posterior probability using the global logistic regression method before determining the local window type and parameters from the variation function. Variation functions were established in four directions to detect anisotropic changes in space. If there are no significant differences among the various directions, a circular local window can be used for ILRBSWT, as shown in Fig. 1; otherwise, an elliptic local window should be used, as shown in Fig. 2. The specific parameters for the local window in the study area were obtained as shown in Fig. 7, and the final local window and

geographical weight attenuation were determined as indicated in Fig. 8 (a) and 8 (b),
respectively.

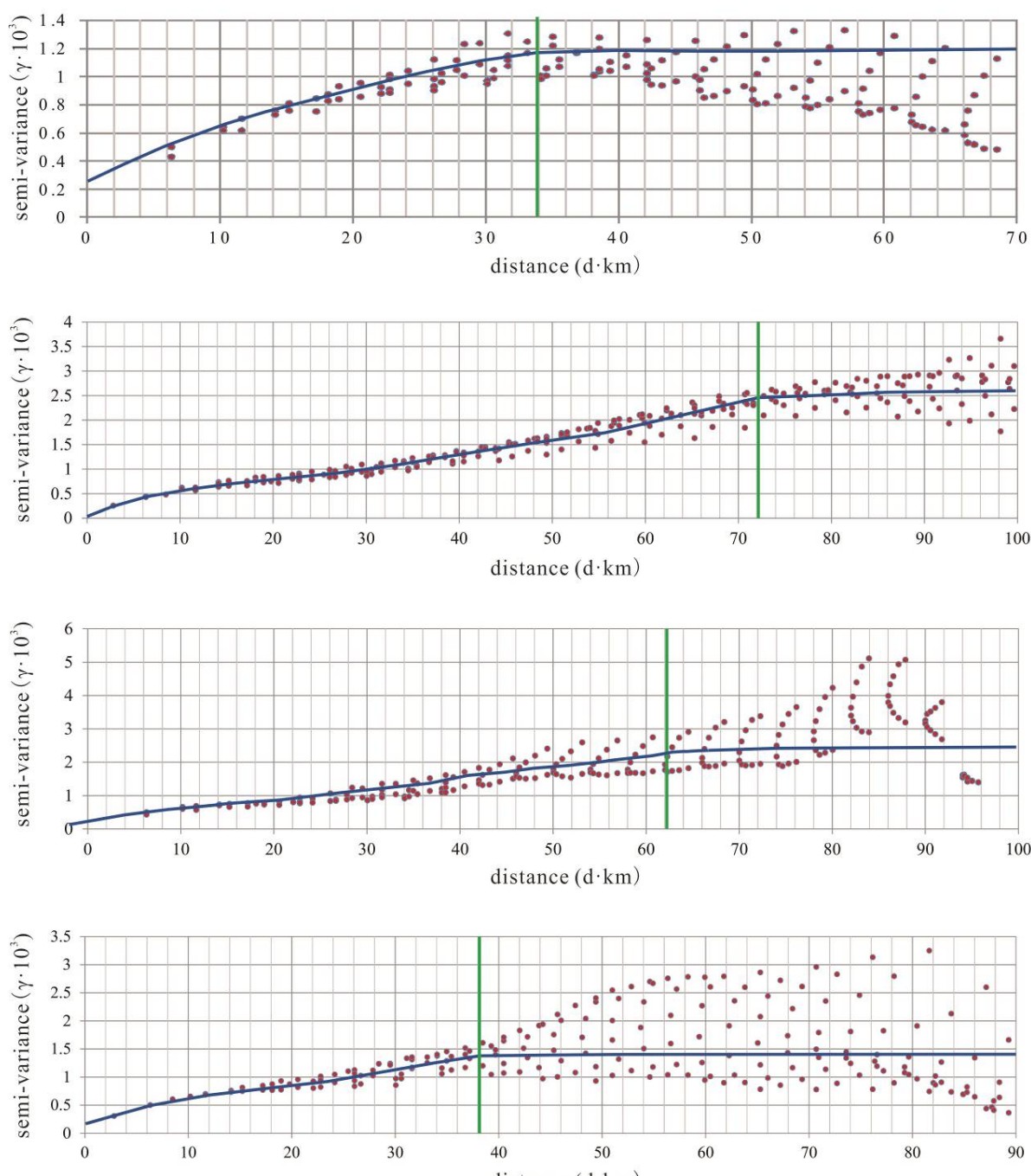


**Fig. 7 Experimental variogram fitting in different directions, where the green lines denote the**
**variable ranges determined for azimuths of (a) 0°, (b) 45°, (c) 90°, and (d) 135°.**

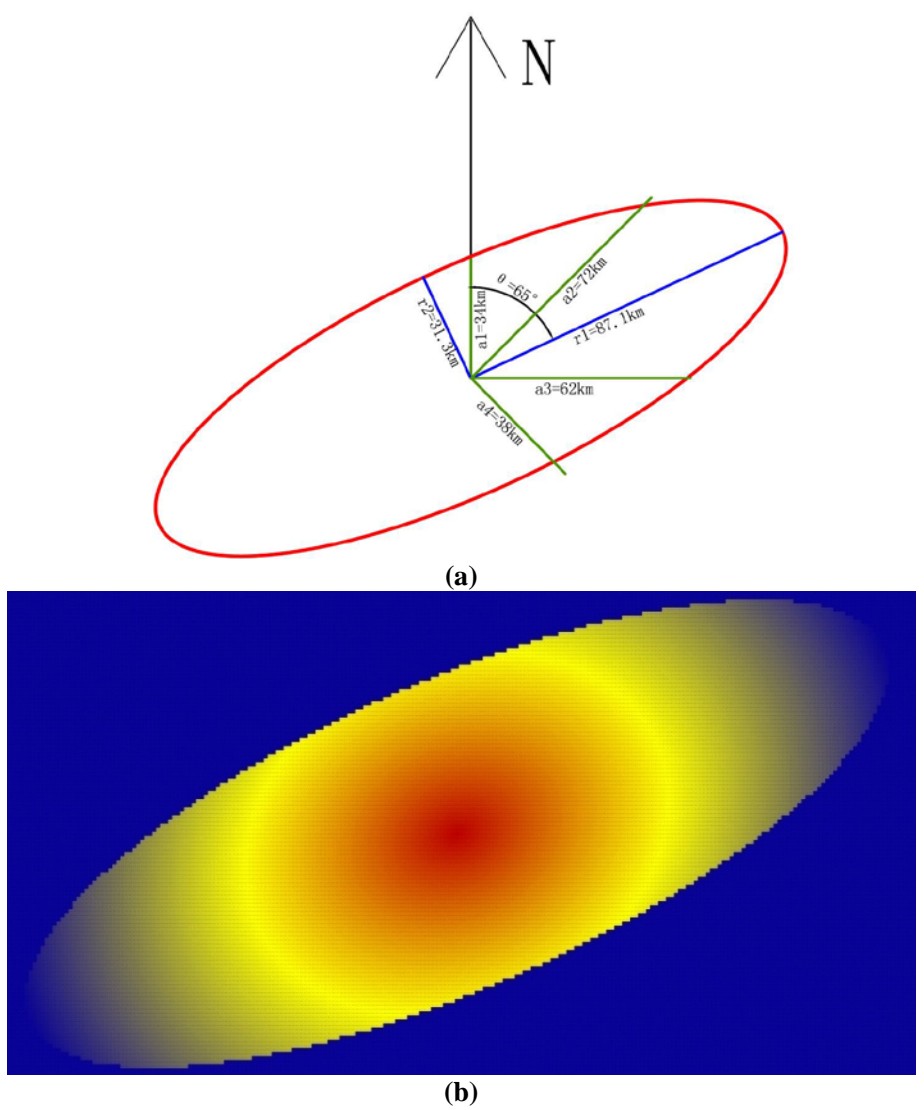

**(a)**

**(b)**


**Fig. 8 Nested spherical model for different directions. The green lines in (a) correspond to those**
**in Fig. 5, and (b) shows the geographical weight template determined based on (a).**
*5.4 Data integration*
Using the algorithm described in section 3.2, ILRBSWT was applied to the study area
according to the parameter settings in Fig. 3. The estimated probability map obtained for Au
deposits by ILRBSWT is shown in Fig. 9 (b), while Fig. 9 (a) presents the results obtained by
logistic regression. As shown in Fig. 8, ILRBSWT better manages missing data than logistic
regression, as the Au deposits in the north part of the study area (with missing data) better fit
within the region with higher posterior probability in Fig. 9 (b) than in Fig. 9 (a).

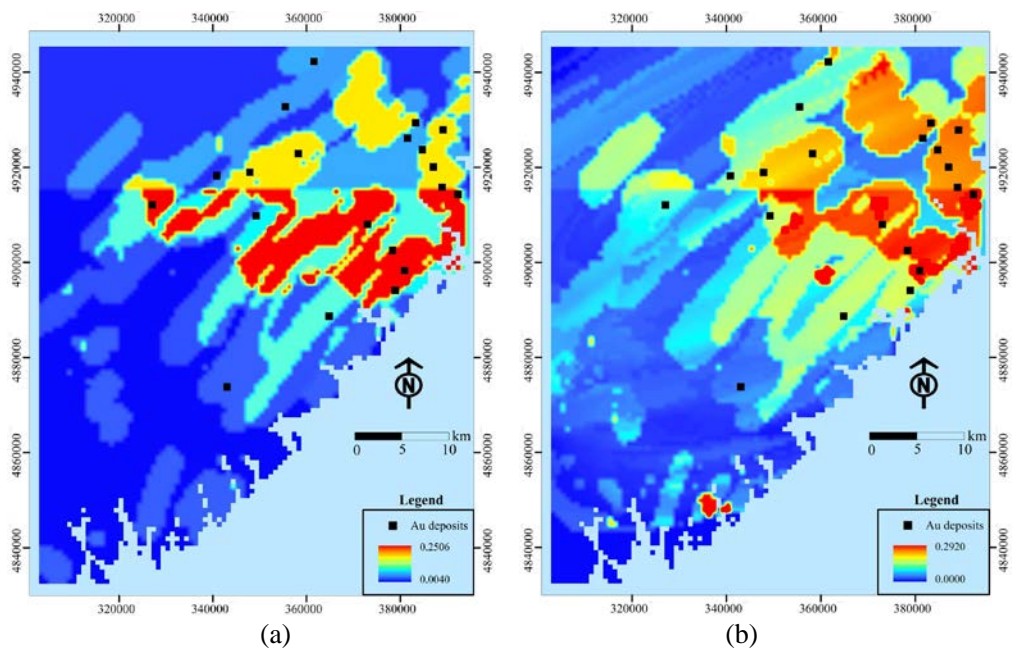

**Fig. 9 Posterior probability maps obtained for Au deposits by (a) logistic regression and (b) ILRBSWT.**

*5.5 Comparison of the mapping results*

To evaluate the predictive capacity of the newly developed and traditional methods, the posterior probability maps obtained through logistic regression and ILRBSWT shown in Fig. 9 (a) and 9 (b) were divided into 20 classes using the quantile method. Prediction-area (P-A) plots (Mihalasky & Bonham-Carter, 2001; Yousefi et al., 2012; Yousefi & Carranza, 2015a) were then made according to the spatial overlay relationships between Au deposits and the two classified posterior probability maps in Fig. 10 (a) and (b) respectively. In a P-A plot, the horizontal ordinate indicates the discretized classes of a map representing the occurrence of deposits. The vertical scales on the left and right sides indicate the percentage of correctly predicted deposits from the total known mineral occurrences and the corresponding percentage of the delineated target area from the total study area (Yousefi & Carranza, 2015a). As shown in Figs. 10 (a) and (b), with the decline of the posterior probability threshold for the mineral occurrence from left to right on the horizontal axis, more known deposits are correctly predicted, and meantime more areas are delimited as the target area; however, the

growth in the prediction rates for deposits and corresponding occupied area are similar before
the intersection point in Fig. 10 (a), while the former shows higher growth rate than the latter
in Fig. 10 (b). This difference suggests that ILRBSWT can predict more known Au deposits
than logistic regression for delineating targets with the same area, and indicates that the
former has a higher prediction efficiency than the latter.

It would be a little inconvenient to consider the ratios of both predicted known deposits

and occupied area. Mihalasky and Bonham-Carter (2001) proposed a normalized density, i.e.
the ratio of the predicted rate of known deposits to its corresponding occupied area. The
intersection point in a P-A plot is the crossing of two curves. The first is fitted from scatter
plots of the class number of the posterior probability map and rate of predicted deposit
occurrences (the "Prediction rate" curves in Fig. 10). The second is fitted according to the
class number of the posterior probability map and corresponding accumulated area rate (the
"Area" curves in Fig. 10). At the interaction point, the sum of the prediction rate and
corresponding occupied area rate is 1; the normalized density at this point is more commonly
used to evaluate the performance of a certain spatial variable in indicating the occurrence of
ore deposits (Yousefi & Carranza, 2015a). The intersection point parameters for both models
are given in Table 1. As shown in the table, 71% of the known deposits are correctly predicted
with 29% of the total study area delineated as target area when the logistic regression is
applied; if ILRBSWT if applied, 74% of the known deposits can be correctly predicted with
only 26% of the total area delineated as the target area. The normalized densities for the
posterior probability maps obtained from the logistic regression and ILRBSWT are 2.45 and
2.85 respectively; the latter performed significantly better than the former.

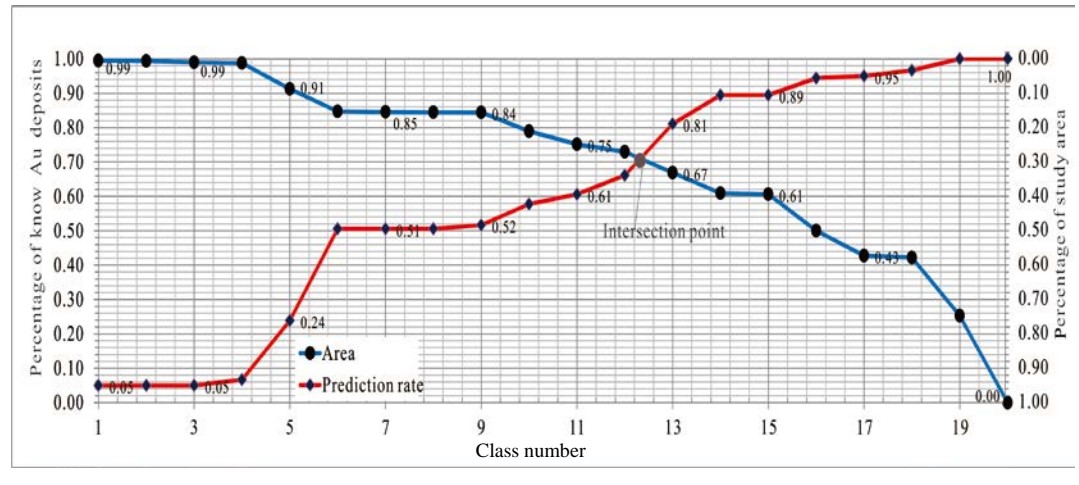

(a)

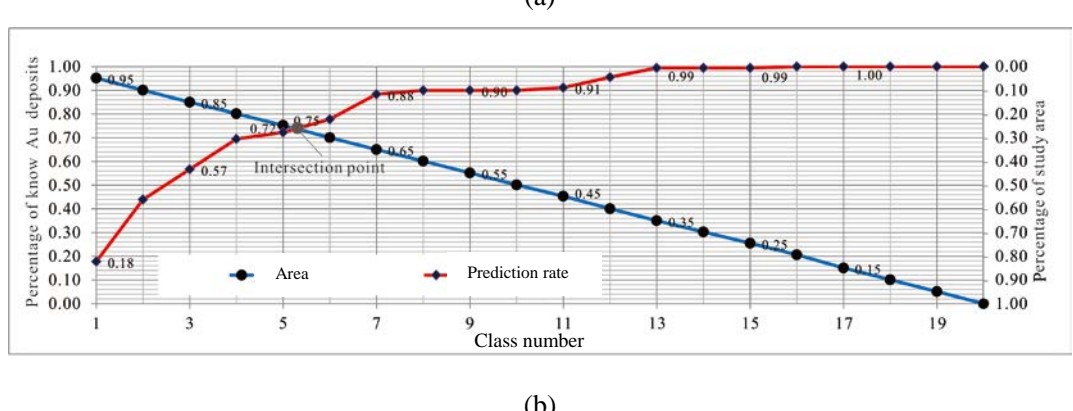

(b)

Fig. 10 Prediction-area (P-A) plots for discretized posterior probability maps obtained by logistic regression and ILRBSWT respectively.

Table 1. Parameters extracted from the intersection points in Figs. 10 (a) and (b).

| Model | Prediction rate | Occupied area | Normalized density |
|---|---|---|---|
| Logistic regression | 0.71 | 0.29 | 2.45 |
| ILRBSWT | 0.74 | 0.26 | 2.85 |

## 6 Discussion

Because of potential spatial heterogeneity, the model parameter estimates obtained based on the total equal-weight samples in the classical regression model may be biased, and they may not be applicable for predicting each local region. Therefore, it is necessary to adopt a local window model to overcome this issue. The presented case study shows that ILRBSWT

can obtain better prediction results than classical logistic regression because of the former's
sliding local window model, and their corresponding intersection point values are 2.85 and
2.45, respectively. However, ILRBSWT has even advantages. For example, characterizing 26%
or 29% of the total study area as promising prospecting targets is too high in terms of
economic considerations. If instead 10% of the total area needs is mapped as the target area,
the proportions of correctly predicted known deposits obtained by ILRBSWT and logistic
regression are 44% and 24%, respectively. The prediction efficiency of the former is 1.8 times
larger than the latter.

In this study, we did not separately consider the influences of spatial heterogeneity,

missing data, and degree of exploration weight all remain, so we cannot evaluate the impact
of each factor. Instead, the main goal of this work was to provide the ILRBSWT tool,
demonstrating its practicality and overall effect. Zhang et al. (2017) applied this model to
mapping intermediate and felsic igneous rocks and proved the effectiveness of the ILRBSWT
tool in overcoming the influence of spatial heterogeneity specifically. In addition, Agterberg
and Bonham-Carter (1999) showed that WofE has the advantage of managing missing data,
and we have taken a similar strategy in ILRBSWT. We did not fully demonstrate the necessity
of using exploration weight in this work, which will be a direction for future research.
However, it will have little influence on the description and application of ILRBSWT tool as
it is not an obligatory factor, and users can individually decide if the exploration weight
should be used.

Similar to WofE and logistic regression, ILRBSWT is a data-driven method, thus it

inevitably suffers the same problems as data-driven methods, e.g., the information loss caused
by data discretization, and exploration bias caused by the training sample location. However,
it should be noted that evidential layers are discretized in each local window instead of the
total study area, which may cause less information loss. This can also be regarded as an
advantage of the ILRBSWT tool. With respect to logistic regression and WofE, some
researchers have proposed solutions to avoid information loss resulting from spatial data
discretization by performing continuous weighting (Pu et al., 2008; Yousefi & Carranza,
2015b, 2015c), and these concepts can be incorporated into further improvements of the
ILRBSWT tool in the future.

**7 Conclusions**
Given the problems in existing MPM models, this research provides an ILRBSWT tool.
We have proven its operability and effectiveness through a case study. This research is also
expected to provide a software tool support for geological exploration researchers and
workers in overcoming the non-stationarity of spatial variables, missing data, and differences
in exploration degree, which should improve the efficiency of MPM work.

*Code availability*
The software tool ILRBSWT v1.0 in this research was developed using C#, and the source
codes and executable programs (software tool) are prepared in the folders "source code for
ILRBSWT in C#" and "Executable Programs for ILRBSWT" respectively. Please find them
in gmd-2017-278-supplement.zip.

*Data availability*
The data used in this research is sourced from the demo data for GeoDAS software
(http://www.yorku.ca/yul/gazette/past/archive/2002/030602/current.htm), which was also
used by Cheng (2008). All spatial layers used in this work are included in the folder "Original
Data" in the format of an ASCII file, which is also found in gmd-2017-278-supplement.zip.

**Acknowledgments**
This study benefited from joint financial support from the Programs of National Natural
Science Foundation of China (Nos. 41602336 and 71503200), China Postdoctoral Science
Foundation (Nos. 2017T100773 and2016M592840), Shaanxi Provincial Natural Science
Foundation (No. 2017JQ7010), and Fundamental Research from Northwest A&F University
in 2017 (No. 2017RWYB08). The first author thanks former supervisor Drs. Qiuming Cheng
and Frits Agterberg for fruitful discussions of spatial weights and providing constructive
suggestions. Great thanks also to the anonymous referees for their helpful suggestions and
corrections.

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
