# Peer review of "An improved logistic regression model based on a spatially weighted technique (ILRBSWT v1.0) and its application to mineral prospectivity mapping"

_Geoscientific Model Development, 2017_

## Referee Comment (RC1) · Anonymous Referee #1 · 19 Jan 2018

1. Line 52-53, these references are so old, please cite more recent referecnes. 2. Line 57-61, it is better to show two recent examples. 3. line 63-67, there are vairious appliactions of GWR in Geosciences, they should be cited here.

---

## Referee Comment (RC2) · Anonymous Referee #2 · 22 Feb 2018

The manuscript presents something that is technically sound. So it can be accepted for publication after addressing the following comments:

The English needs to be improved. It has not been structured well. The statements and propositions have not been organized properly. Reflecting the state of the art is poor as well. The Introduction has not properly been tightened, so the problem and the purpose are not clear.

In Fig. 8, two different data sets were bound together and can explicitly be separated

by a horizontal line. I think there is something wrong. Perhaps it would be better that the two data sets (A and B) be gridded by the same cell size and the spatial values should not be modeled/mapped individually. You should generate a model similar to the Fig. 5.

Weighted evidence layers must be added to the manuscript.

The manuscript presents lack of a Discussion section.

The methods applied, i.e. "weights of evidence" and "logistic regression" are data-driven MPM methods, which carry exploration bias and uncertainty resulting from using classified spatial data and location of known deposits as training sites. Please add a discussion on the disadvantages of such data-driven MPM methods. There are continuous weighting approaches using logistic functions (e.g., logistic-based weighting methods, geometric average function, continuous fuzzification method, and . . .) to avoid the aforementioned uncertainty.

The evaluation method applied could not reflect the efficiency of the two models adequately. So you can see that there is no much difference between the models. I think it would be better if you could apply a prediction-area (P-A) plot and calculate normalized density for the two models to compare them.

The Conclusion is somewhat repetition of the text body. Please re-think about the Conclusion.

Good luck!
* * *

---

## Author Comment (AC1) · 11 Apr 2018

Dear Editor, We appreciate both you and the two anonymous reviewers giving our work (ID: gmd-2017-278) positive comments and giving us the chance to make a further modification of our manuscript. We have carefully modified the manuscript according to the suggestions and comments provided by the reviewers and hope our modification could meet with the requirement of GMD. Attached file includes the responds to the reviewers' suggestions and comments one by one (all suggestions and comments are colored in red, and our proposed changes to the manuscript are

colored blue). At the end of the attachment we provided the comparison between the newest edition and the original edition.

Please also note the supplement to this comment: https://www.geosci-model-dev-discuss.net/gmd-2017-278/gmd-2017-278-AC1-supplement.pdf
* * *
[Figure]

**Supplement:**

Daojun Zhang, Na Ren, and Xianhui Hou

Dear Editor,

We appreciate both you and the two anonymous reviewers giving our work (ID: gmd-2017-278) positive comments and giving us the chance to make a further modification of our manuscript. We have carefully modified the manuscript according to the suggestions and comments provided by the reviewers and hope our modification could meet with the requirement of GMD. Following are the responds to the reviewers' suggestions and comments one by one (all suggestions and comments are colored in red, and our proposed changes to the manuscript are colored blue). At the end of this file we attached the comparison between the newest edition and the original edition.

**Response to Anonymous Referee #1:**

*1. Line 52-53, these references are so old, please cite more recent references.*

Thank you for your comments and suggestions. Here we mainly listed the method research literatures. Unlike application researches, the method researches especially original models (not including modified models) are generally older. Anyway, we have added more recent models here as references, please see lines 57-64 in the comparison edition attached. The new statement is as following.

"(1) Locations are introduced as direct or indirect independent variables. This type of model is still a global model, but space coordinates or distance weights are employed to adjust the regression estimation between the dependent variable and independent variables (Agterberg, 1964; Agterberg and Cabilio, 1969; Agterberg, 1970; Agterberg and Kelly, 1971; Agterberg, 1971; Casetti, 1972; Lesage & Pace, 2009, 2011)."

*2. Line 57-61, it is better to show two recent examples.*

Thank you for your comments and suggestions. We have added more references here, which are about the new applications of models including locations as direct or indirect independent variables, please see lines 64-71 in the comparison edition attached. The new statement is as following.

"For example, Reddy et al. (1991) performed logistic regression by including trend variables for mappingto map the base-metal potential in the Snow Lake area, Manitoba, Canada. In addition, Casetti (1972) developed a ; Helbich & Griffith (2016) compared the spatial expansion method (SEM) to other methods in modeling the house price variation locally, where the regression parameters are themselves functions of the x and y coordinates as well asand their combinations; Yu & Liu (2016) used the spatial lag model (SLM) and spatial error model to control spatial effects in modeling the relationship between $PM_{2.5}$ concentrations and per capita GDP in China."

*3.  Line 63-67, there are various applications of GWR in Geosciences, they should be cited here.*

Thank you for your suggestion and we have added some new literatures bout the application of GWR in different fields here, please see lines 74-78 in the comparison edition attached. The new statementis as following.

"GWR wasmodels were first developed at the end of the 20th century by Brunsdon et al. (1996) and Fotheringham et al. (1996, 1997, 2002) for modeling spatially heterogeneous processes, and it hashave been used widely in the field of geography.geosciences (e.g., Buyantuyev & Wu, 2010; Barbet-Massin et al., 2012; Ma et al., 2014; Brauer et al., 2015)."

**Response to Anonymous Referee #2:**

*The manuscript presents something that is technically sound. So it can be accepted for publication after addressing the following comments:*

1. *The English needs to be improved. It has not been structured well. The statements and propositions have not been organized properly. Reflecting the state of the art is poor as well. The Introduction has not properly been tightened, so the problem and the purpose are not clear.*

Thank you for your suggestions. We have made a major revision to the manuscript. As you can see in the modified manuscript attached, added or subtracted some statements from the original manuscript to clarify the intentions of this work more clearly. We also included the evidential layers in the modified manuscript (please also see Figure R 1). With respect to instruction, we have re-sorted the previous researches in overcoming the non-stationary of spatial variables (especially lines 111-134 in the comparison edition attached), removed the redundant expressions to avoid repetition with later model description parts, and set more natural paragraphs to enhance the level of expression. Some expressions in the summary section have also been modified.

Besides, the English was re-checked thoroughly.

2. *In Fig. 8, two different data sets were bound together and can explicitly be separated by a horizontal line. I think there is something wrong. Perhaps it would be better that the two data sets (A and B) be gridded by the same cell size and the spatial values should not be modeled/mapped individually. You should generate a model similar to the Fig. 5.*

Thank you for your suggestion. We have added that all the raster files in this research are created with the cell size of 1 km x 1 km (lines 481-482 in the comparison edition attached). In fact, it is missing data that caused the sharp differences between the north and south parts (i.e. A and B in Fig. 5) of Fig. 8 (new Fig. 9) rather than data set source, since we have made up a circumstance that there are no geochemical data in region B (lines 485-488 in the comparison edition attached). These expressions are cited following.

"The four independent variables described previously were also used for ILRBSWT modeling in this study (see Figs. 4 (a) to (d)), and they were uniformed in the ArcGIS grid format with a cell size of 1 km × 1 km. To demonstrate the advantages of the new method for missing data processing, we designed an artificial situation in Fig. 5, i.e., grids in region A have values for all four independent variables, while they only have values for two independent variables and no values in the two geochemical variables in region B. "

We acknowledge that the texture looks finer in Fig. 5 (new Fig. 6), and that is because this spatial variable is a continuous variable. However, as a posterior probability layer, Fig. 8 (new Fig. 9) was obtained after the discretizing and integrating the evidence layers, including the buffer layer and the geochemical anomaly layer, which can easily lead to the spatial discontinuity of the grid value. As a result, the texture looks rough, which is not caused by grid size differences.

3. *Weighted evidence layers must be added to the manuscript.*

Thank you for your suggestion and we have accepted it, please see Fig R 4 (Fig. 4 in the attached comparison), which includes all original evidential layers used in this research. Besides, as a sliding window model, ILRBSWT builds predictive model at each local window, and the discretization of original evidential layers and the determination of weights for each class are also based on the local window, thus it is impossible to show the final weights used for modeling.

[Figure]

**Fig. R 1: Evidential layers used to map Au deposits in this study: buffer of anticline axes (a), buffer for the contact of Goldenville–Halifax Formation (b), and background (c) and anomaly (d) separated with the S-A filtering method based on the ore element loadings of the first component.**

4. *The manuscript presents lack of a Discussion section.*

Thank you for your suggestion and we have accepted it. We have added an individual Discussion Section in the new manuscript to discuss the findings and deficiencies of the study (lines 539-602 in the comparison edition attached). Besides, we have added more analyses and discussions in section 5.5 about the comparison of the results of different models (lines 604-638 in the comparison edition attached); please also see details as cited following:

"6 Discussion

Because of potential spatial heterogeneity, the model parameter estimates obtained based on the total equal-weight samples in the classical regression model may be biased, and they may not be applicable for predicting each local region. Therefore, it is necessary to adopt a local window model to overcome this issue. The presented case study shows that ILRBSWT can obtain better prediction results than classical logistic regression because of the former's sliding local window model, and their corresponding intersection point values are 2.85 and 2.45, respectively. However, ILRBSWT has even advantages. For example, characterizing 26% or 29% of the total study area as promising prospecting targets is too high in terms of economic considerations. If instead 10% of the total area needs is mapped as the target area, the proportions of correctly predicted known deposits obtained by ILRBSWT and logistic regression are 44% and 24%, respectively. The prediction efficiency of the former is 1.8 times larger than the latter.

In this study, we did not separately consider the influences of spatial heterogeneity, missing data, and degree of exploration weight all remain, so we cannot evaluate the impact of each factor. Instead, the main goal of this work was to provide the ILRBSWT tool, demonstrating its practicality and overall effect. Zhang et al. (2017) applied this model to mapping intermediate and felsic igneous rocks and proved the effectiveness of the ILRBSWT tool in overcoming the influence of spatial heterogeneity specifically. In addition, Agterberg and Bonham-Carter (1999) showed that WofE has the advantage of managing missing data, and we have taken a similar strategy in ILRBSWT. We did not fully demonstrate the necessity of using exploration weight in this work, which will be a direction for future research. However, it will have little influence on the description and application of ILRBSWT tool as it is not an obligatory factor, and users can individually decide if the exploration weight should be used.

Similar to WofE and logistic regression, ILRBSWT is a data-driven method, thus it inevitably suffers the same problems as data-driven methods, e.g., the information loss caused by data discretization, and exploration bias caused by the training sample location. However, it should be noted that evidential layers are discretized in each local window instead of the total study area, which may cause less information loss. This can also be regarded as an advantage of the ILRBSWT tool. With respect to logistic regression and WofE, some researchers have proposed solutions to avoid information loss resulting from spatial data discretization by performing continuous weighting (Pu et al., 2008; Yousefi & Carranza, 2015b, 2015c, 2016), and these concepts can be incorporated into further improvements of the ILRBSWT tool in the future."

5. *The methods applied, i.e. "weights of evidence" and "logistic regression" are data-driven MPM methods, which carry exploration bias and uncertainty resulting from using classified spatial data and location of known deposits as training sites. Please add a discussion on the disadvantages of such data-driven MPM methods. There are continuous weighting approaches using logistic functions (e.g., logistic-based weighting methods, geometric average function, continuous fuzzification method, and …) to avoid the aforementioned uncertainty.*

Thank you for your suggestions and we have accepted them. We have included in the Discussion Section a description about the shortcomings of the data-driven MPM method, and reviewed previous efforts in overcoming the issues caused by data discretization; please see details in the third paragraph in the discussion section.

6. *The evaluation method applied could not reflect the efficiency of the two models ade-quately. So you can see that there is no much difference between the models. I think it would be better if you could apply a prediction-area (P-A) plot and calculate normalized density for the two models to compare them.*

Thank you for your suggestion, and we have accepted it. We applied the prediction-area (P-A) plot and normalized density in the new manuscript to replace the previous used *t*-value method for model comparison in *"5.5 Comparison of the mapping results"* (lines 538-558 in the comparison edition attached), as is cited following.

"To evaluate the predictive capacity of the newly developed and traditional methods, the posterior probability maps obtained through logistic regression and ILRBSWT shown in Fig. 9 (a) and 9 (b) were divided into 20 classes using the quantile method. Prediction-area (P-A) plots (Mihalasky & Bonham-Carter, 2001; Yousefi et al., 2012; Yousefi & Carranza, 2015a) were then made according to the spatial overlay relationships between Au deposits and the two classified posterior probability maps in Fig. 10 (a) and (b) respectively. In a P-A plot, the horizontal ordinate indicates the discretized classes of a map representing the occurrence of deposits. The vertical scales on the left and right sides indicate the percentage of correctly predicted deposits from the total known mineral occurrences and the corresponding percentage of the delineated target area from the total study area (Yousefi & Carranza, 2015a). As shown in Figs. 10 (a) and (b), with the decline of the posterior probability threshold for the mineral occurrence from left to right on the horizontal axis, more known deposits are correctly predicted, and meantime more areas are delimited as the target area; however, the growth in the prediction rates for deposits and corresponding occupied area are similar before the intersection point in Fig. 10 (a), while the former shows higher growth rate than the latter in Fig. 10 (b). This difference suggests that ILRBSWT can predict more known Au deposits than logistic regression for delineating targets with the same area, and indicates that the former has a higher prediction efficiency than the latter.

It would be a little inconvenient to consider the ratios of both predicted known deposits and occupied area. Mihalasky and Bonham-Carter (2001) proposed a normalized density, i.e. the ratio of the predicted rate of known deposits to its corresponding occupied area. The intersection point in a P-A plot is the crossing of two curves. The first is fitted from scatter plots of the class number of the posterior probability map and rate of predicted deposit occurrences (the "Prediction rate" curves in Fig. 10). The second is fitted according to the class number of the posterior probability map and corresponding accumulated area rate (the "Area" curves in Fig. 10). At the interaction point, the sum of the prediction rate and corresponding occupied area rate is 1; the normalized density at this point is more commonly used to evaluate the performance of a certain spatial variable in indicating the occurrence of ore deposits (Yousefi & Carranza, 2015a). The intersection point parameters for both models are given in Table 1. As shown in the table, 71% of the known deposits are correctly predicted with 29% of the total study area delineated as target area when the logistic regression is applied; if ILRBSWT if applied, 74% of the known deposits can be correctly predicted with only 26% of the total area delineated as the target area. The normalized densities for the posterior probability maps obtained from the logistic regression and ILRBSWT are 2.45 and 2.85 respectively; the latter performed significantly better than the former."

The evaluation results supported the conclusions of this research, Please see Fig. R 2 (Fig. 10 in the comparison edition attached).

[Figure]

(a)

(b)

**Fig. R 2: Prediction-area (P-A) plots for discretized posterior probability maps obtained by logistic regression and** ILRBSWT **respectively.**

7. *The Conclusion is somewhat repetition of the text body. Please re-think about the Conclusion.*

Thank you for your comment and the conclusion has been reorganized:

[revised manuscript text omitted]

---

## Author Response (AR2)

Topical Editor Decision: Publish subject to minor revisions (review by editor) (10 May 2018) by Lutz Gross

Comments to the Author:

Daojun

I am happy to inform that the reviewers are happy with your newest version of the manuscript. To be compliant with the GMD code availability I need to ask you to add the source code to the code provided as supplement.

Thanks.

Lutz Gross

GMD Executive Editor

Dear Lutz,

Thanks for your comments, and we have added the source code with necessary notes. The compressed file of *supplement.zip* has been updated, and the new one includes the original data used in this research, the executable programs for ILRBSWT 1.0 which can be run in the window-based operating systems, and the source code in C#. The source code has been recompiled in Visual Studio 2015 in English version so as to assure that all comments and auto-generates by the system are in the English language.

In the new manuscript, we have checked the references and corrected some errors. Besides, some small modifications were made in the sections of *Code Availability*, *Data Availability*, and *Acknowledgments*.

Best regrards,

Daojun

May 11, 2018

[revised manuscript text omitted]